# Plasma Small Extracellular Vesicles with Complement Alterations in *GRN*/*C9orf72* and Sporadic Frontotemporal Lobar Degeneration

**DOI:** 10.3390/cells11030488

**Published:** 2022-01-30

**Authors:** Sonia Bellini, Claudia Saraceno, Luisa Benussi, Rosanna Squitti, Sara Cimini, Martina Ricci, Laura Canafoglia, Cinzia Coppola, Gianfranco Puoti, Clarissa Ferrari, Antonio Longobardi, Roland Nicsanu, Marta Lombardi, Giulia D’Arrigo, Claudia Verderio, Giuliano Binetti, Giacomina Rossi, Roberta Ghidoni

**Affiliations:** 1Molecular Markers Laboratory, IRCCS Istituto Centro San Giovanni di Dio Fatebenefratelli, 25125 Brescia, Italy; sbellini@fatebenefratelli.eu (S.B.); csaraceno@fatebenefratelli.eu (C.S.); lbenussi@fatebenefratelli.eu (L.B.); rsquitti@fatebenefratelli.eu (R.S.); alongobardi@fatebenefratelli.eu (A.L.); rnicsanu@fatebenefratelli.eu (R.N.); 2Unit of Neurology V—Neuropathology, Fondazione IRCCS Istituto Neurologico Carlo Besta, 20133 Milan, Italy; sara.cimini@istituto-besta.it (S.C.); martina.ricci@istituto-besta.it (M.R.); giacomina.rossi@istituto-besta.it (G.R.); 3Integrated Diagnostics for Epilepsy, Fondazione IRCCS Istituto Neurologico Carlo Besta, 20133 Milan, Italy; laura.canafoglia@istituto-besta.it; 4Department of Advanced Medical and Surgical Sciences, University of Campania “L. Vanvitelli”, 80131 Naples, Italy; cinzia.coppola@unicampania.it (C.C.); gianfranco.puoti@unicampania.it (G.P.); 5Service of Statistics, IRCCS Istituto Centro San Giovanni di Dio Fatebenefratelli, 25125 Brescia, Italy; cferrari@fatebenefratelli.eu; 6CNR Institute of Neuroscience, 20129 Milan, Italy; mrtlombardi@gmail.com (M.L.); giulia.darrigo90@gmail.com (G.D.); c.verderio@in.cnr.it (C.V.); 7MAC-Memory Clinic and Molecular Markers Laboratory, IRCCS Istituto Centro San Giovanni di Dio Fatebenefratelli, 25125 Brescia, Italy; gbinetti@fatebenefratelli.eu

**Keywords:** frontotemporal lobar degeneration, *GRN*, *C9orf72*, extracellular vesicles, endo-lysosomal pathway, plasma, nanoparticle tracking analysis, complement proteins, neuronal ceroid lipofuscinosis, biomarkers

## Abstract

Cutting-edge research suggests endosomal/immune dysregulation in *GRN*/*C9orf72*-associated frontotemporal lobar degeneration (FTLD). In this retrospective study, we investigated plasma small extracellular vesicles (sEVs) and complement proteins in 172 subjects (40 Sporadic FTLD, 40 Intermediate/Pathological *C9orf72* expansion carriers, and 49 Heterozygous/Homozygous *GRN* mutation carriers, 43 controls). Plasma sEVs (concentration, size) were analyzed by nanoparticle tracking analysis; plasma and sEVs C1q, C4, C3 proteins were quantified by multiplex assay. We demonstrated that genetic/sporadic FTLD share lower sEV concentrations and higher sEV sizes. The diagnostic performance of the two most predictive variables (sEV concentration/size ratio) was high (AUC = 0.91, sensitivity 85.3%, specificity 81.4%). C1q, C4, and C3 cargo per sEV is increased in genetic and sporadic FTLD. C4 (cargo per sEV, total sEV concentration) is increased in Sporadic FTLD and reduced in *GRN*+ Homozygous, suggesting its specific unbalance compared with Heterozygous cases. C3 plasma level was increased in genetic vs. sporadic FTLD. Looking at complement protein compartmentalization, in control subjects, the C3 and C4 sEV concentrations were roughly half that in respect to those measured in plasma; interestingly, this compartmentalization was altered in different ways in patients. These results suggest sEVs and complement proteins as potential therapeutic targets to mitigate neurodegeneration in FTLD.

## 1. Introduction

Frontotemporal lobar degeneration (FTLD) is among the most common forms of presenile dementia, mainly affecting people younger than 65, and accounting for up to 15% of all dementias [1,2]. FTLD is typified by degeneration of the frontal and temporal lobes, which results in progressive behavior, personality, executive function, and language deficits. Intracellular ubiquitin and TAR DNA-binding protein (TDP)-43-positive inclusions are the most common pathological hallmarks of both genetic and sporadic FTLD, followed by microtubule-associated protein tau (MAPT) or fused-in-sarcoma (FUS) protein deposition [3].

In general, up to 40% of FTLD patients report a family history of dementia, although only approximately 10% show a clear autosomal dominant trait [4,5,6]. The most acknowledged disease-causing genes are the chromosome 9 open reading frame (*C9orf72*) [7,8], and the genes encoding for progranulin (*GRN*) [9,10], and for microtubule-associated protein tau (*MAPT*) [11,12]. Expansions of a hexanucleotide repeat (GGGGCC) in the first intron/promoter of the *C9orf72* gene are among the most common cause of familial FTLD, accounting for up to 25% of cases [13]. Healthy individuals could carry up to 25 hexanucleotide repeats, with more than 50% carrying only 2 repeats. Pathological expansions (>30 hexanucleotide repeats) typify FTLD [7,8]; however, intermediate expansion (12-30 hexanucleotide repeats) has a risk effect in familial/sporadic FTLD and its repeat unit number influences *C9orf72* expression as well as the disease phenotype in terms of the age of onset and associated clinical subtype [7,13,14]. *GRN* mutations account for around 5% of all FTLD cases and up to 25% of familial FTLD [15]. Among the *GRN* loss of function mutations, the most common are the p.Arg493X (c.1477C > T), identified in the “Mayo Clinic cohort” [16], the Belgian *GRN* p.0 (c.138 + 1G > C) [10,17], and, in Italy, the *GRN* p.Leu271LeufsX10 (c.813-816delCACT), identified in the “Brescia cohort” [18] and the *GRN* Cys157LysfsX97 (c.468-474delCTGCTGT), disclosed in southern Italy [19]. The majority of *GRN* mutations introduce premature termination codons and consequently degrade the mutant RNA by nonsense-mediated decay, causing a reduction in circulating progranulin protein, easily detectable by plasma/serum dosage [20]. Additionally, few missense mutations have been associated with FTLD, and reported to reduce functional progranulin, impairing its secretion or processing [21]. Rare homozygous *GRN* mutations mostly cause neuronal ceroid lipofuscinosis (NCL), a lysosomal storage disorder which shares some neuropathological features with *GRN*-FTLD [22].

Breakthrough molecular studies have suggested lysosomal/endosomal dysfunction, neurotoxic microglia activation, and immune system dysregulation as pathological features in *GRN/C9orf72*-associated FTLD [23,24]: *GRN*-FTLD brains exhibited signs of lysosomal dysfunction such as elevated lipofuscin accumulation, impaired activity of lysosomal enzymes [25], and elevated levels of lysosomal proteins [26]. These findings support the concept of a pathological continuum between *GRN*-FTLD and NCL.

The complement system is a rapid and efficient immune surveillance system; however, its imbalance can contribute to various immune, inflammatory, and neurodegenerative diseases [27]. Loss of *C9orf72* leads to lysosomal accumulation and upregulated immune responses in microglia [24], and to disrupted lysosomal degradation and exocytosis due to impaired autolysosome acidification in macrophages [28]. *C9orf72* deficiency also promotes the transition to an inflammatory condition, characterized by an enhanced type I-interferon signature in microglia, and a complement-mediated cortical synaptic loss, resulting in learning and memory deficits in mice [29]. A deficiency in progranulin was shown to cause excessive microglia activation and a progressive upregulation of lysosomal and innate immunity genes, including increased complement production [23,30]. Notably, in *grn-/-* mice, deletion of the genes encoding for C1qa and C3 mitigates microglial toxicity and recovers TDP-43 proteinopathy and neurodegeneration [30], suggesting complement activation as an important driver of neurodegeneration.

Extracellular vesicles (EVs) are membranous nanoparticles naturally released from cells delimited by a lipid bilayer. EVs are heterogeneous, comprising vesicles of various sizes (small EVs: < 200 nm, medium/large EVs > 200 nm) and of different origin (endosome origin: “exosomes”; plasma-membrane-derived: “ectosomes”) [31,32]. EVs have been demonstrated to play a pivotal role in cell-to-cell communication and molecular transfer (delivering cargo including nucleic acids, lipids, and protein). They also allow cells to reduce the burden of lysosomal storage material or toxic proteins, thus sustaining either protective or pathogenic processes. In patient-derived fibroblasts and induced pluripotent stem-cell-derived motor neurons, *C9orf72* hexanucleotide repeat expansion led to haploinsufficiency and consequent defective intracellular and extracellular vesicle trafficking and a dysfunctional trans-Golgi network phenotype, strengthening the connection between lysosomal imbalance and FTLD [33]. In a previous study of ours, we demonstrated that a null mutation in *GRN* reduces release and alters the exosome composition in primary human fibroblasts [34]. Furthermore, we reported a lower EV concentration and an increased EV size in human plasma samples from patients with Alzheimer’s disease (AD) and related dementias [35]. In line with our results, a reduced exosome concentration in serum/plasma was observed in multiple system atrophy and Parkinson’s disease patients, compared with controls [36]. Increased levels of total Tau, phosphorylated Thr181 Tau, and Aβ42 in neurally derived blood EVs were demonstrated to be an early marker for AD and cognitive decline progression, thus supporting the use of circulating EVs as a sort of “liquid biopsy” [37,38,39]. Thus, the development of advanced platforms able to distinguish EVs subtypes is of great translational value for disease diagnosis and treatment monitoring [40,41].

In the current study, we sought to investigate small EV (sEV) production and complement dysregulation in patients affected by sporadic and genetic FTLD to challenge whether differential molecular pathways are associated with the progressive loss of progranulin or *C9orf72* or whether common mechanisms of immunity/endo-lysosomal pathways can explain pathophysiological processes underlying clinical FTLD subtype overlaps.

## 2. Materials and Methods

### 2.1. Subjects

The project included a total of 172 subjects (129 patients and 43 controls). Patients with sporadic FTLD (n = 40), with FTLD due to *C9orf72* expansion (n = 9 *C9orf72* intermediate expansion carriers, *C9orf72* Int.; n = 31 *C9orf72* pathological expansion carriers, *C9orf72* Pat.), with FTLD due to pathogenic mutations in *GRN* (n = 46 heterozygous mutation carriers, 42 null and 4 missense, *GRN*+ Het.), with NCL due to pathogenic mutations in *GRN* (n = 3 *GRN* homozygous null mutation carriers, *GRN* + Homo.) (Appendix A), and subjects with normal cognitive function as a control group (n = 43, Ctrl), were included in the present retrospective study. Patients were enrolled at the MAC Memory Clinic IRCCS Fatebenefratelli, Brescia, and at the Neurology 5/Neuropathology Unit, IRCCS Besta, Milan. Clinical diagnosis of FTLD was made according to international guidelines [42,43]. A diagnosis of NCL was made by the ultrastructural examination of skin biopsy [22]. Participants signed an informed consent for blood collection and biobanking, as approved by the local ethics committee (approval number 2/1992; 26/2014). *C9orf72* and *GRN* genetic screening was performed previously, as described in [14,15,44]. Demographic and clinical characteristics are reported in Table 1. The study protocol was approved by the local ethics committee (Prot. N. 44/2018).

### 2.2. sEV Isolation and Characterization

sEV isolation was performed with the Total Exosome Isolation Kit (from plasma) (Invitrogen^TM^, Waltham, MA, USA), following the manufacturer’s instructions. Briefly, 125 µL of plasma underwent a two-step centrifugation at room temperature (RT), 2000× *g* for 20 min and 10,000× *g* for 20 min, respectively, to remove cells and debris. The supernatant, transferred into a new tube, was incubated for 10 min at RT with 62.5 µL of 0.2 µM filtered 1X phosphate-buffered saline (PBS) and 37.5 µL of exosome precipitation reagent. The resulting suspension was centrifuged at 10,000× *g* for 5 min. sEV pellets were resuspended with 100 µL of filtered 1X PBS and stored at +4 °C or −20 °C for nanoparticle tracking analysis (NTA). Alternatively, sEV pellets were lysed with 60 µL of ice-cold exosome resuspension buffer (Total Exosome RNA and Protein Isolation Kit, Invitrogen^TM^, Waltham, MA, USA), and stored at −20 °C for Western blotting and biochemical analyses. In parallel, as a negative control, 125 µL of filtered 1X PBS was processed as described above. For sEV characterization, TSG101, CD9, and calnexin expression were analyzed in sEVs by Western blot (Appendix A). Western blotting analysis was performed according to standard protocols. Briefly, resuspended sEVs (5 µL each) were separated using Bolt^TM^ 4–12% Bis-Tris Plus Gels (Invitrogen^TM^, Waltham, MA, USA) with MOPS SDS running buffer (Invitrogen^TM^, Waltham, MA, USA). Samples were electro-transferred onto nitrocellulose membranes (ThermoFisher Scientific, Waltham, MA, USA) for 90 min at 90 V, the membranes were immunoblotted with primary antibodies overnight at + 4 °C (anti-TSG101, anti-CD9, Abcam, Cambridge, UK; anti-Calnexin, BD Biosciences, Franklin Lakes, NJ, USA), and then incubated with horseradish peroxidase-conjugated secondary antibodies (Invitrogen^TM^, Waltham, MA, USA) for 1 h at +37 °C. Immuno-positive bands were detected by ultra-sensitive enhanced chemiluminescence (ThermoFisher Scientific, Waltham, MA, USA) according to the manufacturer’s instructions.

### 2.3. Nanoparticle Tracking Analysis

sEVs resuspended in PBS from patients and controls were analyzed with the Nano-Sight NS300 Instrument (Malvern, Worcestershire, UK). Samples were diluted with 0.2 µm filtered 1X PBS to obtain an optimal range of 20–150 particles/frame. For each sample, 5 60-second videos were recorded and data were processed using NanoSight NTA Software 3.2. Optimized post-acquisition settings were kept constant during the analysis of all samples. Data obtained were the particle concentration (particles/mL) and average size (nm). Raw concentration data (particles/mL) from NTA were normalized to obtain EV concentrations in the human plasma sample. The negative control included did not interfere with the sample analysis because it was below the detection limit.

### 2.4. Biochemical Analyses

C1q, C4, and C3, either as soluble complement proteins in plasma or as protein cargo in circulating sEVs, were assessed using the Human Complement Magnetic Bead Panel 2—MILLIPLEX^®^ MAP Kit (Millipore, Billerica, MA, USA), following the manufacturer’s procedures. Briefly, after rinsing the plate with wash buffer, 25 µL of sEVs resuspended in exosome resuspension buffer (dilution 1:20,000 in assay buffer) or 25 µL of diluted plasma (dilution 1:40,000 in assay buffer) was added to each well, together with 25 µL of fluorescently labeled capture antibody-coated beads. The plate was shaken overnight at 4 °C in the dark, then washed using a handheld magnet and incubated for 1 h at RT with 50 µL of biotinylated detection antibodies for each well. Afterwards, 50 µL of streptavidin–phycoerythrin was added into each well and incubated for 30 min at RT. The samples were then washed again, resuspended with sheath fluid, and read on the Bio-Plex 200 system (BioRad, Hercules, CA, USA). The median fluorescent intensity was analyzed using 5-parameter logistic curve-fitting to calculate the concentrations of C1q, C4, and C3 in the samples. sEV-associated complement protein concentration data were normalized or not on sEV concentration (C1q, C4, C3 concentrations per sEV, ng/sEV; total sEV C1q, C4, C3 concentrations, ng/mL, respectively). All analyses were performed in duplicate.

### 2.5. Statistical Analysis

Normality assumptions of continuous variables were assessed by Kolmogorov–Smirnov tests. One-way ANOVAs, with Bonferroni post hoc tests, were used for comparisons within the groups of normally distributed demographic variables. Kruskal–Wallis tests, with Dunn’s post hoc tests, were used for comparisons of non-normally distributed demographic variables. Linear (for normally distributed variables) and generalized linear (for non-normally distributed variables) models were performed for comparisons across the groups (adjusting for age). Chi-squared tests (or Fisher’s exact tests when appropriate) were used to assess the association between categorical variables and the group variable. A machine learning technique, classification tree (CT) [45], was applied to detect the best (in terms of classification performance) predictors for discriminating controls versus patient group. In detail, CT was carried out on 2 categories: Ctrl vs. all patients (*C9orf72* Int., *C9orf72* Pat., *GRN* + Het., *GRN* + Homo., Sporadic FTLD) as a categorical dependent variable depending on the following quantitative covariates: sEV concentration, sEV size, C1q/C4/C3 concentration per sEV. Different tree-growing algorithms (CHAID, exhaustive CHAID and classification and regression trees (CART)) were applied in order to obtain the CTs that best fit the data and provided the best predictive performance. The output of the CTs was given by different classification pathways (defined by estimated covariate cut-offs), and for each of them, the probability of the most likely diagnostic group is provided. Finally, the diagnostic performance of the best variables (in terms of discrimination performance found by the CTs) was assessed by the area under the curve (AUC) obtained by receiver operating characteristics (ROC). Comparisons of the AUC were assessed with the DeLong test. All analyses were performed with SPSS software and significance was set at 0.05.

## 3. Results

### 3.1. Plasma sEVs Are Altered in FTLD Patients

Isolated plasma EVs were CD9+ (a tetraspanin), TSG101+ (a cytosolic protein recovered in EVs), and calnexin (endoplasmic reticulum residential protein, absent in EVs) (Appendix A), with a size <200 nm (“small EVs”). Plasma sEV concentrations were decreased in all patient groups compared with the Ctrl (*p* < 0.001, generalized linear model with pairwise comparison), even though not reaching the statistical threshold in the *C9orf72* Int. group (Table 1 and Figure 1a). sEV size was significantly increased in all groups compared with Ctrl, except for the *GRN*+ Homo. Furthermore, sEV size was significantly increased in *GRN*+ Het. compared with *C9orf72* Pat. and Sporadic FTLD groups (*p* < 0.001, linear model with pairwise comparison; Table 1 and Figure 1b). The ratio between sEV concentration and size (sEV conc./sEV size) was lower in all patient groups compared with the Ctrl (*p* < 0.001, generalized linear model with pairwise comparison; Figure 1c).

### 3.2. C1q, C4, and C3 Dysregulation in sEVs and Plasma of FTLD Patients

The study of plasma sEV complement protein cargoes (ng/sEV) revealed significant increases in C1q, C4, and C3 in *C9orf72* Pat., *GRN*+ Het., and Sporadic FTLD patients compared with the Ctrl (Table 1 and Figure 2a). In addition, the C4 sEV cargo (ng/sEV) was significantly (i) reduced in *GRN*+ Homo. compared with *GRN*+ Het., *C9orf72* Pat., and Sporadic FTLD patients and (ii) increased in Sporadic FTLD compared with *GRN*+ Het. (Table 1 and Figure 2a, *p* < 0.001, generalized linear model with pairwise comparison). Considering the total sEV C1q, C4 and C3 concentration (ng/mL, not normalized values), we found a significant reduction in C4 in *GRN*+ Homo with respect to all groups, and a significant increase in Sporadic FTLD compared with *GRN*+ Het. (*p* < 0.001, generalized linear model with pairwise comparison; Figure 2b). Furthermore, the C1q, C4, and C3 protein levels were also analyzed in plasma. As shown in Figure 2c and Table 1, only C3 was altered in FTLD, highlighting a different trend in genetic versus sporadic FTLD (with a significant increase in plasma C3 in *C9orf72* Pat. and *GRN*+ Het. compared with Sporadic FTLD; *p* < 0.001, generalized linear model with pairwise comparison). Moreover, we evaluated the ratio between C1q, C4, and C3 protein levels in EVs and plasma. As shown in Figure 2d, we observed (i) a significant increase in C4 sEVs/plasma ratio in Sporadic FTLD compared with *GRN*+ Het. and Homo. (*p* < 0.001, generalized linear model with pairwise comparison); (ii) a significant decrease in C4 in *GRN*+ Homo. compared with all groups (FTLD patients and controls); (iii) a significant increase in C3 sEVs/plasma ratio in Sporadic FTLD compared with the Ctrl and all other patient groups (FTLD and NCL patients) (*p* < 0.001, generalized linear model with pairwise comparison). C1q analysis did not show differences among the groups.

### 3.3. sEV Concentration and Size Discriminate Ctrl from Patients

The considered socio-demographic variables were not associated with the diagnostic categories; therefore, the sole sEV variables (sEV concentration, sEV size, C1q, C4, and C3 sEV concentration) were included as covariates in the CT run considering the dichotomized group variables Ctrl vs. the whole patient group. The overall classification performance of the model had a correct classification percentage of 90.7%. The discrimination between Ctrl and patients was driven by sEV size and sEV concentration. Specifically, the sEV size alone (with values > 114.54 nm) could distinguish the majority of patients from Ctrl (88.5% vs. 11.5%). Moreover, among the subjects with sEV size > 114.54 nm, those with an sEV concentration ≤ 2.3 × 10^11^ sEVs/mL were classified as patients with a very high percentage (94.5%, Figure 3a). Finally, to estimate the diagnostic performance of sEV concentration/sEV size ratio, ROC analyses were performed (Figure 3b). The reliability of the sEV concentration:sEV size ratio to discriminate patients from the Ctrl was high (AUC = 0.91), with a sensitivity of 85.3% and specificity of 81.4%, considering the cut-off point of 1.29 × 10^9^. ROC curves for *C9orf72* Int., *C9orf72* Pat., *GRN*+ (Het. and Homo.), and Sporadic FTLD vs. Ctrl had a similar AUC (0.77, 0.93, 0.94, 0.90, respectively), with DeLong test *p*-values for AUC comparisons all larger than 0.087.

## 4. Discussion

The main result of the current study is that genetic or sporadic forms of FTLD shared lower sEV concentrations and higher sEV sizes than healthy controls. Independently from the cause of FTLD onset—loss of progranulin, *C9orf72* gene expansion or pathological processes underlying Sporadic FTLD—a dysregulation of sEV production seems to be a common mechanism underlying the disease process. This new concept is supported by our investigation encompassing the assessment of plasma sEV indices in a relatively large group of FTLD patients, including sporadic patients, genetic patients (intermediate/pathological *C9orf72* expansion carriers, *GRN* heterozygous mutation carriers), and *GRN* homozygous mutation carriers affected by NCL, a lysosomal storage disorder. Compared with controls, in all patient groups, we observed: (i) a reduction in sEV concentration, excluding the *C9orf72* Int. carriers; and (ii) an increase in the sEV size, excluding the *GRN*+ Homo. group that, given the very rare frequency of this mutation in homozygosis, was composed of a very low number of carriers.

We previously demonstrated, in human primary fibroblasts from patients, that the loss of *GRN* results in reduced exosome secretion and altered exosome composition [34]. Herein, we demonstrated that these alterations are also detectable in the blood; furthermore, they are common to either the *C9orf72* or Sporadic FTLD.

As discussed above, another important parameter is the size, the most predictive variable able to discriminate patients (88.5%) from controls in the classification tree, followed by sEV concentration (94.5%). Overall, the model based on sEV indices correctly classified 90.7% of patients and healthy controls. The diagnostic performance of the two most predictive variables (as combined index sEV concentration/sEV size ratio) was high (AUC = 0.91), with a sensitivity of 85.3% and specificity of 81.4%, considering the cut-off point of 1.29 × 10^9^ of the sEV concentration/sEV size ratio in discriminating patients from healthy controls. Of note, we previously demonstrated the potential of this combined index in detecting not only FTLD, but also AD and dementia, with Lewy bodies (DLB) patients with respect to controls [35], suggesting that an alteration of intercellular communication mediated by EVs might be a common molecular pathway underlying neurodegenerative dementias.

The main result from the complement system-related protein study is that C1q, C4, and C3 cargo per sEV is increased in *C9orf72* Pat., *GRN*+ Het., and Sporadic FTLD patients compared with Ctrl, suggesting an alteration of sEV complement cargo in genetic and sporadic FTLD. Moreover, the increased C4 cargo per sEV in Sporadic FTLD compared with *GRN*+ Het./Homo. suggests a specific C4 activation in sEVs from Sporadic FTLD patients. Conversely, *GRN*+ Homo. was characterized by a significant reduction in C4 cargo per sEV with respect to the *GRN* + Het. and *C9orf72* Pat. groups, thus suggesting an “allele dose-related effect” (homozygous versus heterozygous) and/or an alteration related to different clinical phenotypes (NCL versus FTLD). The number of sEVs is reduced in patients; therefore, it is also important to evaluate the total amount of sEV complement proteins: interestingly, the total sEV C1q, C4, and C3 concentration was not altered in all groups except for Sporadic FTLD, with a significant increase in total C4 sEV concentration, and for *GRN*+ Homo., with a significant decrease in total C4 sEV concentration, further suggesting the specific imbalance of this protein in Sporadic FTLD and *GRN*+ Homo. with an opposite direction and compared with *GRN*+ Het. cases. It is important to note that *GRN*+ Homo. is associated with a different clinical phenotype, the storage lysosomal disorder (NCL), affecting *GRN* homozygous mutation carriers.

Additionally, analysis of circulating plasma complement only highlighted C3 alterations, and specifically, higher levels of soluble C3 in genetic FTLD (*C9orf72* Pat. and *GRN*+ Het.) compared with Sporadic FTLD. Exploring complement protein compartmentalization, in controls, the concentrations of C3 and C4 detected in sEVs were roughly half that with respect to those measured in plasma. Statistical analysis of C1q, C4, C3 sEVs/plasma concentration showed a differential (i) C4 compartmentalization in Sporadic FTLD compared with *GRN*+ Het./Homo.; (ii) C4 compartmentalization in *GRN*+ Homo. compared with all groups (FTLD patients and controls); (iii) C3 compartmentalization in Sporadic FTLD with respect to *GRN*+ Het./Homo., *C9orf72* Int./Pat. and controls: of note, in Sporadic FTLD, we observed a more than twofold enrichment of this protein in sEVs than in plasma. These data were driven for C4 by the selective enrichment of C4 sEV cargo in Sporadic FTLD and by very low levels of C4 sEV cargo in *GRN*+ Homo. NCL patients; for C3, by reduced levels in plasma of Sporadic FTLD compared with the genetic FTLD forms.

Previous studies described higher cerebrospinal fluid levels of C3b and C1q in symptomatic carriers of mutations in *GRN*, *C9orf72*, or *MAPT* compared with pre-symptomatic carriers and healthy non-carriers, suggesting a role of these proteins as biomarkers of disease progression [46]. Interestingly, mouse studies showed that (i) progranulin deficiency promotes synaptic pruning by microglia via complement activation [23], and (ii) neurotoxicity in the microglia of *GRN* knockout mice can be mitigated by deleting the genes encoding C1q and C3 or by blocking complement-mediated formation of the membrane attack complex [30]. An increase in C3 and a decrease in C1s in the serum of sporadic FTLD patients compared with healthy controls was recently reported [47,48].

To the best of our knowledge, this is the first study reporting an alteration in plasma sEV-associated complement proteins in sporadic and genetic FTLD patients. Our results found similarities with studies on AD patients reporting higher levels of classical and alternative complement pathway proteins in enriched populations of astrocyte-derived exosomes obtained from human plasma [49]. Moreover, preclinical models of AD showed that C1q levels increased in neuronal and astrocytic extracellular vesicles (NEVs and AEVs) isolated from plasma with respect to controls and positively correlated with C1q from the cortex and hippocampus [50]. In line with these data, we described an increased C1q EV cargo in genetic and sporadic FTLD compared with controls. Altogether, our data as well as the literature data support a role of EV complement proteins in neurodegenerative dementias, including genetic FTLD forms.

Emerging data argue for an interdependence between the production of EVs and endosomal pathway in the brain [51]. Herein, we demonstrated an alteration of blood sEV parameters and composition in genetic and sporadic FTLD. We recently reported that genes controlling key endo-lysosomal processes (i.e., protein sorting/transport, clathrin-coated vesicles uncoating, lysosomal enzymatic activity regulation) might be involved in neurodegenerative dementias, including FTLD [52]. How these alterations impact EV release/composition, brain–periphery communication, and neurodegeneration is still a matter of discussion. CNS-derived EVs have been shown to cross the blood–brain barrier into the bloodstream; therefore, they have attracted substantial attention as sources of biomarkers for various neurodegenerative diseases because they can be isolated via a minimally invasive blood-draw to assess the biochemical status of the CNS [53].

A limitation of the study is the small number of patients included in the *C9orf72* Int. and *GRN*+ Homo. groups. Despite these limitations, we achieved the result of a significant decrease in sEV concentration in *GRN*+ Homo. compared with Ctrl, likely associated with the lack of progranulin. However, we acknowledge that the limited number of patients in these groups could have also masked other significant differences among all patients. Confirmatory studies in larger sample populations, as well as studies employing different technologies for EV quantification/characterization, are warranted.

In conclusion, our study suggests that altered sEVs mediated cell-to-cell communication and complement release might be a common molecular pathway in genetic and sporadic FTLD. These results emphasize the importance of EVs and complement in FTLD and support the feasibility of developing strategies that target EVs and complement proteins as potential therapeutic targets to mitigate neurodegeneration in FTLD.

## Figures and Tables

**Figure 1 cells-11-00488-f001:**
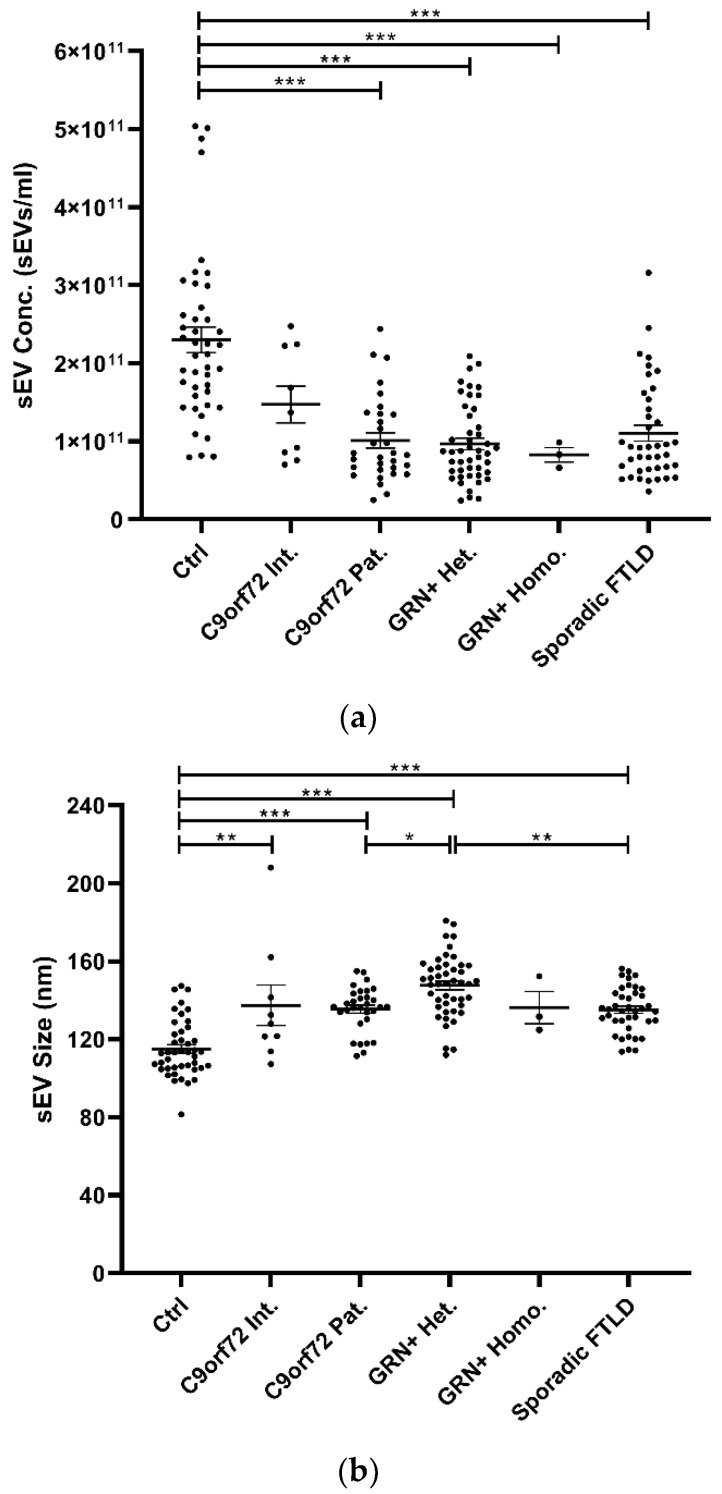
Analysis of plasma sEV concentration and size by NTA in patient and control groups. (**a**) sEV concentration (sEVs/mL), (**b**) sEV size (nm), and (**c**) sEV concentration/sEV size ratio in Ctrl, *C9orf72* Int., *C9orf72* Pat., *GRN*+ Het., *GRN*+ Homo., and Sporadic FTLD groups. Mean ± SEM; * *p* < 0.05, ** *p* < 0.01, *** *p* < 0.001 obtained by generalized linear model (adjusted for age) for non-normally distributed variables (sEV concentration and sEV concentration/sEV size ratio) or by linear model (adjusted for age) for normally distributed variable (sEV size).

**Figure 2 cells-11-00488-f002:**
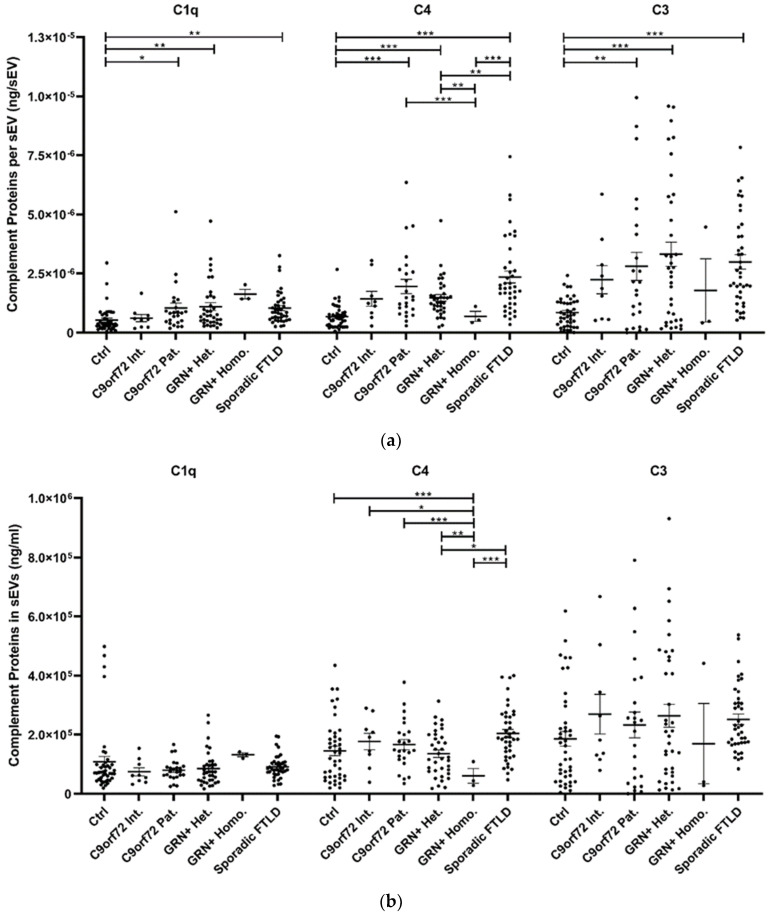
C1q, C4, and C3 proteins (**a**) concentration per sEV (ng/sEV); (**b**) total concentration in sEVs (ng/mL), not normalized values; (**c**) concentration in plasma (ng/mL); and (**d**) total complement protein concentration in sEVs (ng/mL)/complement protein concentration in plasma (ng/mL) ratio in Ctrl, *C9orf72* Int., *C9orf72* Pat., *GRN*+ Het., *GRN*+ Homo., and Sporadic FTLD. Mean ± SEM; * *p* < 0.05, ** *p* < 0.01, *** *p* < 0.001 obtained by the generalized linear model (adjusted for age) for non-normally distributed variables.

**Figure 3 cells-11-00488-f003:**
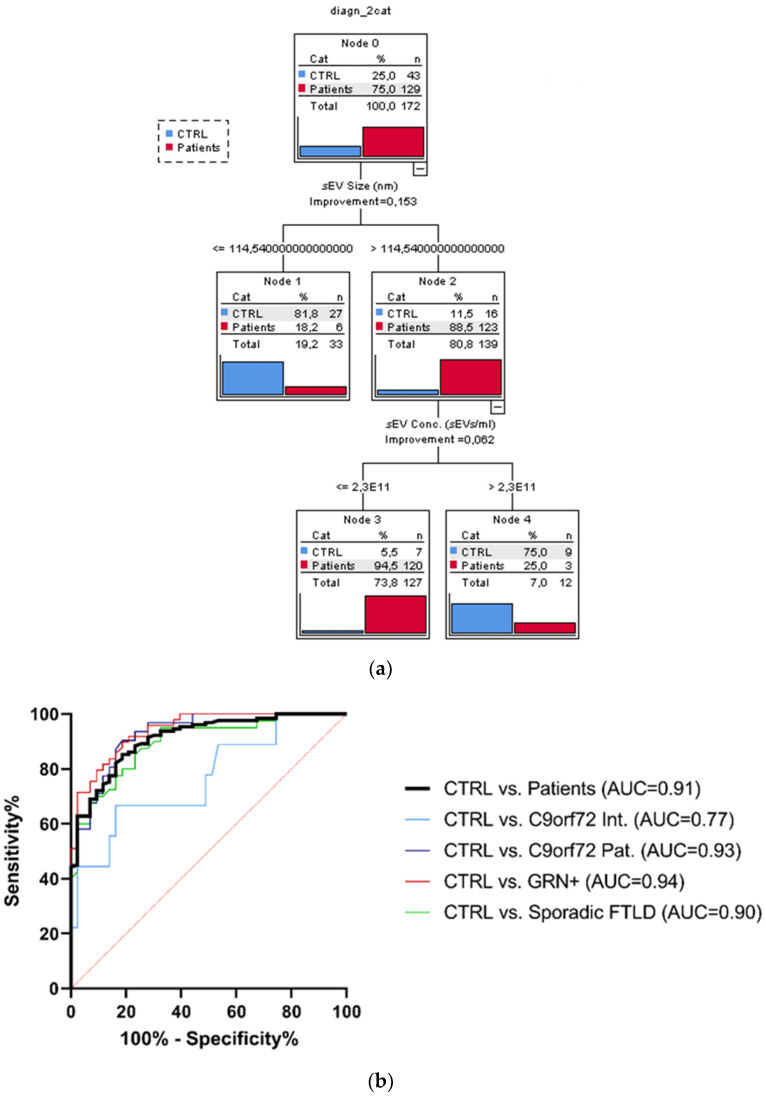
Classification tree and ROC curves. (**a**) Classification obtained with dichotomized group variables, controls (Ctrl) vs. all patients (*C9orf72* Int., *C9orf72* Pat., *GRN*+ Het., *GRN*+ Homo., and Sporadic FTLD), based on the most predictive variables: sEV concentration and size, out of all biological variables associated with the groups. (**b**) Diagnostic capacity of sEV concentration/sEV size ratio to discriminate Ctrl from all patients. AUC comparison with DeLong test, ns *p* > 0.087.

**Table 1 cells-11-00488-t001:** Clinical, demographic, and biological characteristics of patients and controls included in the study.

	Ctrl	*C9orf72* Int.	*C9orf72* Pat.	*GRN* + Het	*GRN* + Homo	Sporadic FTLD	*p*-Value
N.	43	9	31	46	3	40	
Sex (% female)	72.1	44.4	48.4	45.7	33.3	65.0	0.069 ^a^
Age, years	68.0 ± 10.6	67.2 ± 5.2	62.3 ± 9.9	61.4 ± 8.2	31.3 ± 6.8	67.7 ± 10.4	<0.001 ^b^
Onset, years	/	64.1 ± 6.2	60.5 ± 9.9	59.8 ± 8.4	23.3 ± 1.5	64.4 ± 10.8	<0.001 ^b^
Education, years	11.0 ± 3.7	8.6 ± 4.9	7.4 ± 2.9	7.9 ± 4.1	n.a.	8.0 ± 4.6	<0.05 ^c^
sEV Conc., sEVs/mL	2.3 × 10^11^ ± 1.1 × 10^11^	1.5 × 10^11^ ± 7.1 × 10^10^	1.0 × 10^11^ ± 5.4 × 10^10^	9.7 × 10^10^ ± 4.9 × 10^10^	8.3 × 10^10^ ± 1.6 × 10^10^	1.1 × 10^11^ ± 6.3 × 10^10^	<0.001 ^d^
sEV Size, nm	115.1 ± 14.5	137.4 ± 31.0	135.5 ± 11.5	147.6 ± 15.7	136.3 ± 14.3	135.1 ± 11.6	<0.001 ^e^
C1q, ng/sEV	5.3 × 10^−7^ ± 5.4 × 10^−7^	6.2 × 10^−7^ ± 4.5 × 10^−7^	1.0 × 10^−6^ ± 1.0 × 10^−6^	1.1 × 10^−6^ ± 1.0 × 10^−6^	1.6 × 10^−6^ ± 3.4 × 10^−7^	1.0 × 10^−6^ ± 6.8 × 10^−7^	<0.001 ^d^
C4, ng/sEV	6.7 × 10^−7^ ± 4.9 × 10^−7^	1.4 × 10^−6^ ± 9.5 × 10^−7^	2.0 × 10^−6^ ± 1.5 × 10^−6^	1.5 × 10^−6^ ± 8.3 × 10^−7^	6.9 × 10^−7^ ± 3.7 × 10^−7^	2.4 × 10^−6^ ± 1.6 × 10^−6^	<0.001 ^d^
C3, ng/sEV	8.5 × 10^−7^ ± 6.2 × 10^−7^	2.2 × 10^−6^ ± 1.8 × 10^−6^	2.8 × 10^−6^ ± 2.9 × 10^−6^	3.3 × 10^−6^ ± 3.0 × 10^−6^	1.8 × 10^−6^ ± 2.3 × 10^−6^	3.0 × 10^−6^ ± 1.9 × 10^−6^	<0.001 ^d^
C1q Plasma, ng/mL	9.6 × 10^4^ ± 2.5 × 10^4^	1.0 × 10^5^ ± 2.7 × 10^4^	7.9 × 10^4^ ± 1.7 × 10^4^	8.3 × 10^4^ ± 3.0 × 10^4^	7.0 × 10^4^ ± 1.8 × 10^3^	9.4 × 10^4^ ± 2.6 × 10^4^	0.439 ^d^
C4 Plasma, ng/mL	4.2 × 10^5^ ± 1.6 × 10^5^	4.6 × 10^5^ ± 3.5 × 10^5^	3.9 × 10^5^ ± 1.4 × 10^5^	4.7 × 10^5^ ± 1.4 × 10^5^	3.7 × 10^5^ ± 1.4 × 10^5^	3.7 × 10^5^ ± 1.2 × 10^5^	0.072 ^d^
C3 Plasma, ng/mL	5.3 × 10^5^ ± 5.5 × 10^5^	8.6 × 10^5^ ± 5.6 × 10^5^	1.3 × 10^6^ ± 6.5 × 10^5^	1.1 × 10^6^ ± 9.0 × 10^5^	8.9 × 10^5^ ± 1.1 × 10^6^	3.2 × 10^5^ ± 4.1 × 10^5^	<0.001 ^d^

Ctrl = controls; *C9orf72* Int. = *C9orf72* intermediate expansion carriers; *C9orf72* Pat. = *C9orf72* pathological expansion carriers; *GRN*+ Het. = *GRN* heterozygous null and missense mutation carriers; *GRN*+ Homo. = *GRN* homozygous null mutation carriers; Sporadic FTLD = patients with sporadic frontotemporal lobar degeneration. ^a^ Chi-squared test (or Fisher’s exact test when appropriate); ^b^ one-way ANOVA test; ^c^ Kruskal–Wallis test; ^d^ generalized linear model (adjusted for age); ^e^ linear model (adjusted for age). Data are presented as the mean ± standard deviation.

## Data Availability

The data presented in this study are openly available in the Mendeley Data Repository at doi:10.17632/kds9sb4z6t.2 [54].

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
