# Peer review of "Plasma Small Extracellular Vesicles with Complement Alterations in *GRN*/*C9orf72* and Sporadic Frontotemporal Lobar Degeneration"

_cells, 2022, doi:10.3390/cells11030488_

Round 1
Reviewer 1 Report
This is a unique study and has been carefully done. It is also very well written.
The details of methods and statistics described in detail.
The limitation of small number of c9orf mutation cases adequately described.
Reviewer 2 Report
The authors propose to use the complement protein modification (abundance) to be used as a potential biomarkers in Frontotemporal lobar degenerescence in sporadic and in GRN/C9orf72 forms.
The article needs to be clearly modified in deep adding some data to ensure that the EVs used are EVs, such as electron microscopy or related to visualize the EVs, identification of small EV markers, ...
In the present form, this paper is not acceptable for publication. Please find below my recommendations:
- Introduction:
- A deeper discussion on EV and exosomes (origin, size ..) can be appreciable, as well as some precaution using the term 'exosome' since the proposed characterization can't ensure that the used EVs are exosomes. Please take a look at these references: Kowal et al., 2016 (PNAS); Mathieu et al., 2021 (Nat. Comm); Théry et al., 2018 (JEV).
- In addition to the lower concentration of small EVs, it has been describe a clear modification of EV contents in neurodegenerative diseases such as Alzheimer's disease (AD). This argument would reinforce the interest of discriminating the complement molecules on EVs in FLTD.
- Figures: the overall figures are very small, and therefore not so easy to follow for the readers. Some rectangles around each figure A, B, C and D are also visible in Figure 2. Please optimize the figures size/disposition.
- Results:
- Excepting complement proteins, nothing can figure out that the EV isolated are so called "exosomes". Without any further characterization, please refer to as small EVs in agreement with the Minimum Instructions for the Study of EVs (MISEV, Lötvall et al., 2014; Théry et al., 2018).
- Statistics are described as a One-Way ANOVA with a Bonferroni post-hoc test, but according to the data, the significance is discutable.
- Differences observed in Figure 3b are very thin, and have to be taken into account to minimize the message of a potential kind of biomarker.
- How could you explain that the plasma EV concentration is lower in sick patients groups compared to the control?
Reviewer 3 Report
The current manuscript by Bellini et al, entitled ‘Plasma Extracellular Vesicles with Complement Alterations in GRN/C9orf72 and Sporadic Frontotemporal Lobar Degeneration’ investigated EVs production and complement dysregulation in patients affected by sporadic and genetic FTLD to challenge whether differential molecular pathways do associate with the progressive loss of progranulin or C9orf72 or whether common mechanisms of immunity/endo-lysosomal pathways can explain pathophysiological processes underlying clinical FTLD subtypes overlap.
Overall, the manuscript is well written and the subject matter falls under the scope of the journal but the study falls short in the research planning/presentation.
Below are my comments:
NTA: Can the author mention how many total particles were counted/sample? Were there any set points like a minimum of 1000 completed tracks? Also, mention the instrument setting used to eliminate background noise. Do you have representative videos for each group?
The NTA measures only particle sizes/concentrations and doesn't identify any particular vesicles types. Also, it has limitations. [https://www.aquiles.me/limitations_of_nanoparticle_tracking_analysis/]
Line 208: Plasma EV concentration was decreased in all patient groups compared to Ctrl…How old the patients’ samples are compared to the control samples? Larger vesicles tend to rupture/get damaged easily compared to smaller vesicles. Long storage significantly degrades larger EVs thereby reducing total EV numbers.
I would like to see the concentration of EVs by measuring/comparing EV markers for e.g., CD9/CD81/Tsg101 ELISA.
Indirect measurement of EV concentration by using a methodology that analyzes general “particle concentrations” is not recommended when separating two disease groups based on the analyzed data.
See the published manuscript by Yan et al, 2019 that showed the differences between EVs when two isolation methods were used and the size/concentration analysis by DLS and TRPS. Will you be able to discuss in brief how using a PEG-based kit to isolate EV might have limited the sensitivity of the analyzed data? Cite this study and/or any similar publications in the appropriate section.
Line 210-211: EV size was significantly increased in all groups compared to Ctrl, except for the GRN+ Homo. Furthermore, EV size was significantly increased in GRN+ Het. compared to C9orf72 Pat. and Sporadic FTLD groups…Provide representative TEM images from control and patient groups, with a uniform scale bar.
Also, discuss why could be the reason for seeing the differences in EV concentration and size in different groups.
Line 169-170: Briefly, after rinsing the plate with wash buffer, 25 μl of EVs resuspended in Exosome Resuspension Buffer (dilution 1:20,000 in Assay Buffer) or 25 μl of diluted plasma (dilution 1:40,000 in Assay Buffer) were added in each well, together with 25 μl of fluorescently-labeled capture antibody-coated beads…the plasma was crude plasma, right? Not the EV deplete plasma? If so, then EVs in the diluted plasma also contributed to the obtained data?
Figure 2, labels ‘a’ and ‘b’ are missing. Also, if possible, make the label's fonts slightly bigger.
Line 266-269: The discrimination between Ctrl and patients was driven by EV size and EV concentration. Specifically, the EV size alone (with values > 114.54 nm) could classify the majority of patients from Ctrl (88.5% vs. 11.5%)…But the methods used to analyze the concentration and size are not confirmatory. This has been done solely by using the NTA. I understand that several studies have used NTA, and while it provides some basic information on the analyzed particles but doesn’t provide any information on EVs. These data must be accompanied by other markers of EV. This is the USP and also the drawback of the study. Also, the authors need to show representative EV characterization data.
Round 2
Reviewer 2 Report
The revised version of the proposed article looks better in a general point of view. Some points have to be precise before accepting it for publication :
- Since the authors defined the small EVs as vesicles with a diameter inferior to 200nm, and since they used such vesicles all along the manuscript, they have to precise it of refer to these vesicle as small EVs as well as in the title. The general name 'EVs' in the title suggests that the study focused also on the medium/large EVs but it's not.
- Figure 2.C and 2.D are missing ...
- Suplementary Figure 1: two of the potential small EV marker are expressed in the samples extracted, and seems to have a decreased expression after mutations. Can the authors add a negative marker such as calnexin or alpha-actinin4 to be sure of the 'purity' of their small EVs?
Reviewer 3 Report
After the revision, the overall acceptability of the current manuscript has been improved significantly. The authors have changed/updated the information suggested. Some sentences that required proper clarifications have now been addressed. Some critical information related to EV characterization has also been updated. The addition of the new data will help readers understand the scientific matter presented, in a clear way. The manuscript now appears much matured and can be considered for publication based on the comments/suggestions made by other reviewers.
Below are a couple of minor points that requires change/addition
Line: 110-117: Furthermore, we reported a lower EV concentration and an increased EV size in human plasma samples from patients with Alzheimer's disease (AD) and related dementias [35]....It will be relevant here to mention another recent finding that showed a decrease in total plasma exosome concentration from the control group to the PD and MSA groups, which help aid the separation of these groups from each other [doi: 10.1007/s00401-021-02324-0] PMID: 33991233
Line 404-405: How these alterations impact on EV release/composition, brain-periphery communication and neurodegeneration is still matter of discussion....delete 'on'...add 'a matter'...Also, Hornung et al in a review article have summarized several aspects of the advantages and challenges of CNS originating EVs, authors may add the reference here. [doi: 10.3389/fnmol.2020.00038], PMID: 32265650
